## RESEARCH ARTICLE

# Identification and validation of an interaction between the E3 ubiquitin ligase WWP1 and the Transcriptional Co-Activator WBP2 in the human heart

Meaghan E. Arnold[1], Ymani Wright[1], Natalie K. Grantham[2], Douglas L. Pittman[2] and Lydia E. Matesic[1,*]

## ABSTRACT

Impaired proteostasis is a cellular hallmark of aging, and the ubiquitin-proteasome system is a fundamental driver of proteostasis. As an E3 ubiquitin ligase, WW-domain containing protein 1 (WWP1) expression and activity are tightly regulated in cells, while its deregulation has been described in cancer, in neurodegenerative diseases, and in heart failure. However, the protein–protein interaction network of WWP1 is understudied, particularly in the heart. Here, we conducted a yeast-two hybrid (Y2H) screen of a human heart library and identified 21 putative WWP1 interactors, including 12 whose expression and potential function in the heart were previously unappreciated. Central in the identified protein–protein interaction network was WBP2 (WW domain binding protein 2), an oncogenic transcriptional co-activator. Utilizing immunofluorescence and proximity ligation assays, it was confirmed that endogenous WWP1 can co-localize and interact with WBP2 in human heart tissue, and, using the Y2H system, we showed that this interaction is dependent upon the associations between WW domains 1 and 3 from WWP1 and PY domains 2 and 3 of WBP2. In total, these data serve as a launching pad to identify broader protein networks regulated by WWP1 and the regions of interaction which might be targetable to reduce hallmarks of cellular aging.

KEY WORDS: WWP1, WBP2, Protein–protein interaction, Yeast-two hybrid, Cardiac aging

## INTRODUCTION

One of the key tenets of cellular existence is the maintenance of the protein content, a phenomenon known as proteostasis. As cells age, there is a loss of this cellular hallmark, as evidenced by accumulation of both intracellular and extracellular protein aggregates, which underlies certain degenerative pathologies like Alzheimer's, Parkinson's, or cardiac amyloidosis (Bart et al., 2025; Lopez-Otin et al., 2013, 2023; Lopez-Otin and Kroemer, 2021). The ubiquitin-proteasome system is the first line of defense for clearing misfolded and aggregated proteins from many tissues, including heart muscle, and is an important part of the cellular machinery responsible for protein quality control. With chronic stress brought on by aging or disease states, the capacity of this machinery can be overwhelmed, compromising protein integrity and, eventually, resulting in cell death (Kobak et al., 2025; Mainali et al., 2023).

As an E3 ubiquitin ligase, WW-domain containing protein 1 (WWP1) plays an important role in proteostasis. In conjunction with an E1 (ubiquitin activating enzyme) and an E2 (ubiquitin conjugating enzyme), WWP1 binds to specific targets in the cell and transfers one or more ubiquitin moieties to a lysine residue within the substrate protein via a covalent linkage. The biological consequences of this post-translational modification vary from altered target protein function and intracellular localization to degradation by the proteasomal or endolysosomal machinery (Damgaard, 2021). Because WWP1 can target and regulate a myriad of substrates, its expression, activity, and turnover are tightly controlled to prevent inappropriate ubiquitination and consequent cellular dysfunction (Behera and Reddy, 2023). Expression of WWP1 is deregulated with age, and reports show decreased expression of WWP1 is associated with replicative senescence in human fibroblasts (Cao et al., 2011), while overexpression has been described in age-related diseases like various cancers (recently reviewed in Hu et al., 2021; Kuang et al., 2021; Lei et al., 2024), murine models of age-related bone loss (Shu et al., 2013), and heart failure (Zhao et al., 2021). Interestingly, germline variants in *WWP1* have recently been associated with normocephalic autism spectrum disorder (Novelli et al., 2020) and its expression increases with the expression of mutant huntingtin, contributing to aggregate formation and cellular toxicity (Lin et al., 2016), suggesting that homeostatic levels of WWP1 are essential to overall protein quality control, particularly in post-mitotic cell populations.

The predominant isoform of the human WWP1 protein is 922 amino acids long and is composed of a number of functional domains, including an N-terminal C2 domain, which mediates association with lipid bilayers in response to changes in intracellular calcium (Nalefski and Falke, 1996; Plant et al., 1997), four tandem WW domains, which are involved in protein-protein interactions, particularly with proline-rich target regions (Sudol et al., 1995a,b; Sudol and Hunter, 2000), and, at its very C-terminus, there is an enzymatic HECT (homologous to E6-AP carboxy terminus) domain that covalently binds ubiquitin and transfers it to a target protein (Verdecia et al., 2003). It has been noted that the enzymatic activity of WWP1 can be fine-tuned through intramolecular interactions between the WW domains and the HECT domain which inhibit its ubiquitin ligase activity (Wang et al., 2019). A similar autoregulatory mechanism has been described in the paralogous proteins ITCH and WWP2 [as well as their *Drosophila* ortholog Su(dx)], and these can be relieved by binding to adaptor proteins or by protein phosphorylation (Chen et al., 2017; Shah and Kumar, 2021; Yao et al., 2018; Zhu et al., 2017).

[1]Department of Biological Sciences, University of South Carolina, Columbia, SC 29208, USA. [2]Department of Drug Discovery and Biomedical Sciences, College of Pharmacy, University of South Carolina, Columbia, SC 29208, USA.

*Author for correspondence (lmatesic@biol.sc.edu)

D.L.P., 0000-0001-9038-2150; L.E.M., 0000-0002-6460-2321

Biology Open

In the mouse, we have shown that myocardial expression of *Wwp1* increases with age, and global overexpression of WWP1 promotes left ventricular hypertrophy, arrhythmias, interstitial fibrosis, and diastolic dysfunction – phenotypes consistent with cardiac aging (Basheer et al., 2015; Matesic et al., 2020). Further, deletion of *Wwp1* was shown to be protective against the negative sequalae of left ventricular pressure overload in mice, suggesting that WWP1-regulated pathways might serve as viable therapeutic targets in cardiac dysfunction (Snyder et al., 2021; Zhao et al., 2021). However, the protein targets of WWP1 in the heart are vastly understudied. Here, we conducted a yeast-two hybrid (Y2H) screen of a human heart library and identified 21 putative WWP1 interactors, including 12 whose expression and potential function in the heart were previously unappreciated. Of particular interest was WBP2 (WW domain binding protein 2), a transcriptional co-activator and theorized oncoprotein (Sabbaghian et al., 2025; Tabatabaeian et al., 2020), similar to WWP1. Utilizing immunofluorescence and proximity ligation assay (PLA), it was confirmed that a subset of the endogenous WWP1 protein pool co-localizes and interacts with WBP2 in human heart tissue. Using the Y2H system, we show that this interaction is dependent upon the interactions between WW domains 1 and 3 from WWP1 and PY domains 2 and 3 of WBP2. In total, these data serve as a launching pad to identify broader protein networks regulated by WWP1, which might be targetable to reduce hallmarks of cellular aging.

## RESULTS

### Y2H screen identifies 21 putative WWP1 interactors in the human heart

The longest open reading frame (ORF) of human WWP1 was cloned in frame into the pGBK-T7 yeast expression vector, producing an N-terminal fusion of amino acids 1-147 of the GAL4 DNA binding domain with WWP1. The sequence-verified construct was transformed into AH109 haploid cells using the Frozen-EZ yeast transformation II kit (Zymo). Transformants were selected on synthetic defined (SD) media lacking tryptophan (SD-Trp). At least five independent clones were assessed for autoactivation of the *MEL1*, *HIS3*, and *ADE2* reporter genes by plating on SD-Trp plates supplemented with X-α-gal (5-Bromo-4-chloro-3-indolyl α-D-galactopyranoside), on SD-His plates, and on SD-Ade plates, respectively. Consistent with lack of autoactivation, the patches grown from pGBK-WWP1 transformed AH109 colonies were white on SD-Trp plates containing X-α-gal and were not capable of growth on either SD-His or SD-Ade plates (data not shown).

Since there was no autoactivation of the GAL4 reporter genes, the AH109 yeast containing the pGBK-WWP1 construct was used as bait to screen a human heart cDNA library (Takara) using a mating approach. Resulting diploids were plated on quadruple drop out (QDO) media (i.e. synthetic defined media lacking tryptophan, leucine, histidine, and adenine), to stringently select for those cells possessing both the pGBK 'bait' and pGAD 'prey' plasmids well as expression of the *HIS3* and *ADE2* reporter genes. There were 21 interactors identified (Table 1), and approximately half of these were isolated from at least three positive yeast colonies. WWP1 and all the putative interactors were analyzed using STRING v12.0 (Szklarczyk et al., 2023). The resulting network (Fig. 1) had 11 edges and a protein-protein interaction (PPI) enrichment p-value of $3.06 \times 10^{-5}$, meaning that this set of proteins has more interactions among themselves than would be expected for a set of proteins of the same size and degree distribution randomly selected from the human genome. Eight of the proteins (red, Fig. 1) are reported to

**Table 1. WWP1 interactors identified in the Y2H screen**

| Gene symbol | Description | Number of positive yeast colonies obtained | PY motif(s)? |
|---|---|---|---|
| MT-CO1 | Mitochondrial cytochrome c oxidase subunit 1 | 15 | Yes |
| MYBPC3 | Cardiac myosin-binding protein C | 14 | Yes |
| TXNIP | Thioredoxin-interacting protein | 11 | Yes |
| WBP2 | WW domain-binding protein 2 | 6 | Yes |
| SYNPO2 | Synaptopodin 2 | 5 | Yes |
| NEXN | Nexillin F-actin binding protein | 4 | No |
| MT-ND1 | Mitochondrial NADH dehydrogenase subunit 1 | 4 | No |
| DAZAP2 | DAZ-associated protein 2 | 3 | Yes |
| LITAF | Lipopolysaccharide-induced tumor necrosis factor-alpha (TNFα) factor | 3 | Yes |
| NREP | Neuronal regeneration-related protein | 2 | No |
| SHISA4 | SHISA family member 4 | 2 | Yes |
| EWSR1 | Ewing sarcoma (EWS) breakpoint region RNA-binding protein 1 | 1 | Yes |
| PSMF1 | Proteasome inhibitor subunit 1 | 1 | Yes |
| CPSF6 | Cleavage and polyadenylation specificity factor 6 | 1 | Yes |
| WBP1 | WW domain-binding protein 1 | 1 | Yes |
| DGCR2 | DiGeorge syndrome critical region gene 2 | 1 | Yes |
| MT-CYB | Mitochondrial cytochrome B subunit of complex III | 1 | No |
| SF3B4 | Splicing factor 3b, subunit 4 | 1 | Yes |
| LAPTM5 | Lysosomal-associated multi-spanning membrane protein5 | 1 | Yes |
| LRPAP1 | Low density lipoprotein receptor-related protein associated protein 1 | 1 | No |
| PDLIM7 | PDZ and LIM domain protein 7 | 1 | Yes |

have high expression in the human heart while five (purple, Fig. 1) have been associated with heart disease (both with a false discovery rate stringency of less than 5%). Interestingly, the central edge (between WWP1 and WBP2), with a confidence score of 0.824 (where a high confidence interaction score is defined as 0.700 corresponding to a false discovery rate stringency of less than 1%), did not feature in known examples of cardiac expression nor in cardiomyopathy. Therefore, this putative interaction piqued our interest and was explored further.

### Validation of the WWP1–WBP2 interaction

Although the feasibility of this interaction is substantiated by the fact that multiple different cDNA clones of WBP2 were pulled out of the library as well as the fact that previous Y2H (Lim et al., 2016), affinity chromatography and mass spectroscopy (Huttlin et al., 2021; Nielsen et al., 2019), and *in vitro* enzymatic studies (French et al., 2017) have shown that the two proteins can physically interact, we sought to further validate our initial findings and determine whether this interaction occurs endogenously in the heart. To accomplish this, we first cloned the 786 bp ORF of WBP2 into both the pGBK-T7 and pGAD-T7 expression vectors and used these to test the WWP1–WBP2 interaction in both orientations in the Y2H system. Following sequence confirmation of successful cloning,

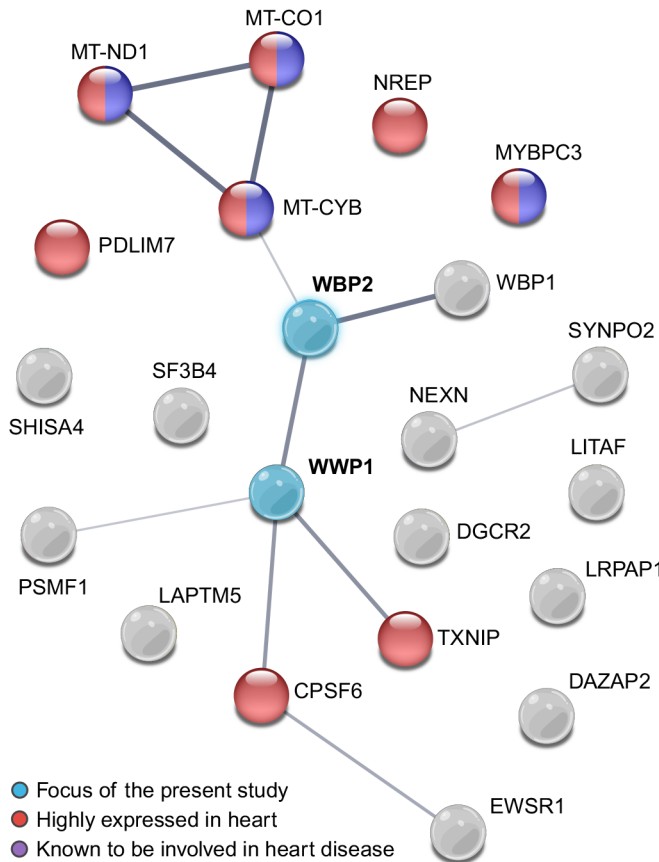

**Fig. 1. STRING PPI network identifies significant connections between putative WWP1 interactors.** WWP1, along with the 21 putative interactors identified from the Y2H screen of the human heart library, were entered into STRING v12.0. The full STRING network for both functional and physical protein associations in *Homo sapiens* is shown here with each protein being represented as a bubble and thickness of the line demarcating edges (*n*=11) reflecting the confidence in strength of support. The minimum threshold for interaction was medium confidence (0.400), corresponding to a false discovery rate of <5% (i.e. alpha-value <0.05). The PPI enrichment *P*-value was 3.06×10$^{-5}$. Functional enrichments included expression in heart (false discovery rate *P*-value=0.0020), shown in red, and association with heart disease (false discovery rate *P*-value=0.0169), colored purple.

pGAD-WWP1 and pGBK-WBP2 or pGAD-WBP2 and pGBK-WWP1 were co-transformed into haploid AH109 cells and plated on synthetic defined media lacking both tryptophan and leucine (double drop out, DDO) to select for cells that had taken up both plasmids. Independent transformants were selected to characterize the expression of GAL4-dependent reporter genes on QDO media by serial dilution assay as well as to quantify strength of interaction using a β-galactosidase assay.

As seen in Fig. 2A, irrespective of orientation, when both WWP1 and WBP2 were present in the same cell, the yeast was capable of growth on QDO plates even when diluted 1:1000 or 1:10,000. In contrast, if AH109 yeast were co-transformed with one empty vector (pGAD-T7 or pGBK-T7) and either WWP1 or WBP2, no growth was evidenced on QDO plates, similar to the negative control (AH109 co-transformed with pGAD-T7 and pGBK-T7). This experiment additionally confirmed that neither protein had autoactivation capacity in either expression vector. The WWP1-WBP2 interaction in the Y2H system was corroborated by quantitative β-galactosidase assay (Fig. 2B). Specifically, while there was no statistically significant difference in β-galactosidase

activity between pGAD-WWP1/pGBK-WBP2 versus pGAD-WBP2/pGBK-WWP1 co-transformed AH109 cells, both of these displayed significantly more activity than any of the other bait/prey combinations.

To elucidate whether there is endogenous co-localization of WWP1 and WBP2 in the heart, co-immunofluorescence was performed on formalin-fixed, paraffin-embedded sections of normal human ventricular tissue. While overall staining for WBP2 was weak, there was detectable punctate WBP2 signal on the cytoplasmic face of the sarcolemma of some cardiomyocytes (green, Fig. 3A). This partially overlapped the punctate cytoplasmic WWP1 signal seen in the cardiomyocytes (purple, Fig. 3A), particularly at the presumptive cell membrane (white, Fig. 3A). Importantly, the no primary negative control sample, when imaged with the same exposure, brightness, and contrast settings displayed minimal non-specific background staining associated with tyramide signal amplification of WBP2 (Fig. 3A, bottom row). Due to low levels of endogenous expression of WBP2 protein, we employed PLA to visualize potential rare PPI within the human myocardium. Using the same antibodies as for immunofluorescence, we were able to detect distinct PLA complex formation (small red puncta, Fig. 3B) in tissue sections derived from two independent donors. To resolve whether this represented a statistically significant increase from the autofluorescent signal emitted from erythrocytes in the negative control (in which one or both of the primary antibodies was omitted), we quantified both the area of PLA signal per field as well as the number of particles in images acquired from five random 40x fields from two different sections for all samples using ImageJ. All fields were subject to the same auto thresholding and masking. This analysis revealed that while, on average, 38.400±18.277 (s.d.) particles were observed in fields from the negative control sections, a statistically significant increase in particles was seen in the sections where both mouse anti-WWP1 and rabbit anti-WBP2 primary antibodies were applied (97.900±28.099, *P*<0.0001 by unpaired, two-tailed *t*-test, Fig. 3C). Similarly, the mean area of PLA signal per field was significantly higher in the experimental cohort than when no primaries were added (766.790±381.367 versus 363.299±185.493, *P*<0.001 by Mann–Whitney *U*-test, Fig. 3C). Because of the paucity of readily available and independent commercially available specimens for analysis, we additionally sought to corroborate these findings through scRNA-seq data contained in the Heart Cell Atlas v2, comprised of data obtained from 25 adult donors aged 20-75 years old (Kanemaru et al., 2023). In querying the expression of *WWP1* and *WBP2* in just ventricular myocytes (*n*=190,710), we found that the majority of cells or nuclei modestly expressed both genes, though the expression of WWP1 was generally higher (Fig. 3D), similar to what we had observed in our immunofluorescence, thus providing additional confirmation of our findings.

## WW domains 1 and 3 from WWP1 interact with PY domains 2 and 3 of WBP2

Subsequently, we sought to determine the domains of WWP1 and WBP2 that mediated this PPI. Because WW domains are known to have affinity for proline-rich sequences like PY motifs (i.e. a Pro-Pro-X-Tyr sequence), we used site-directed mutagenesis to target the second proline in each of the three PY motifs (amino acid residues 168, 198, and 250) in WBP2 and substituted with an alanine, as this has been shown to abrogate binding to WW domains without impacting the overall stability of the protein (Chan et al., 2011; Chen and Sudol, 1995). These mutations were performed singly (yielding mutants mutPY1, mutPY2, or mutPY3) or in

Biology Open

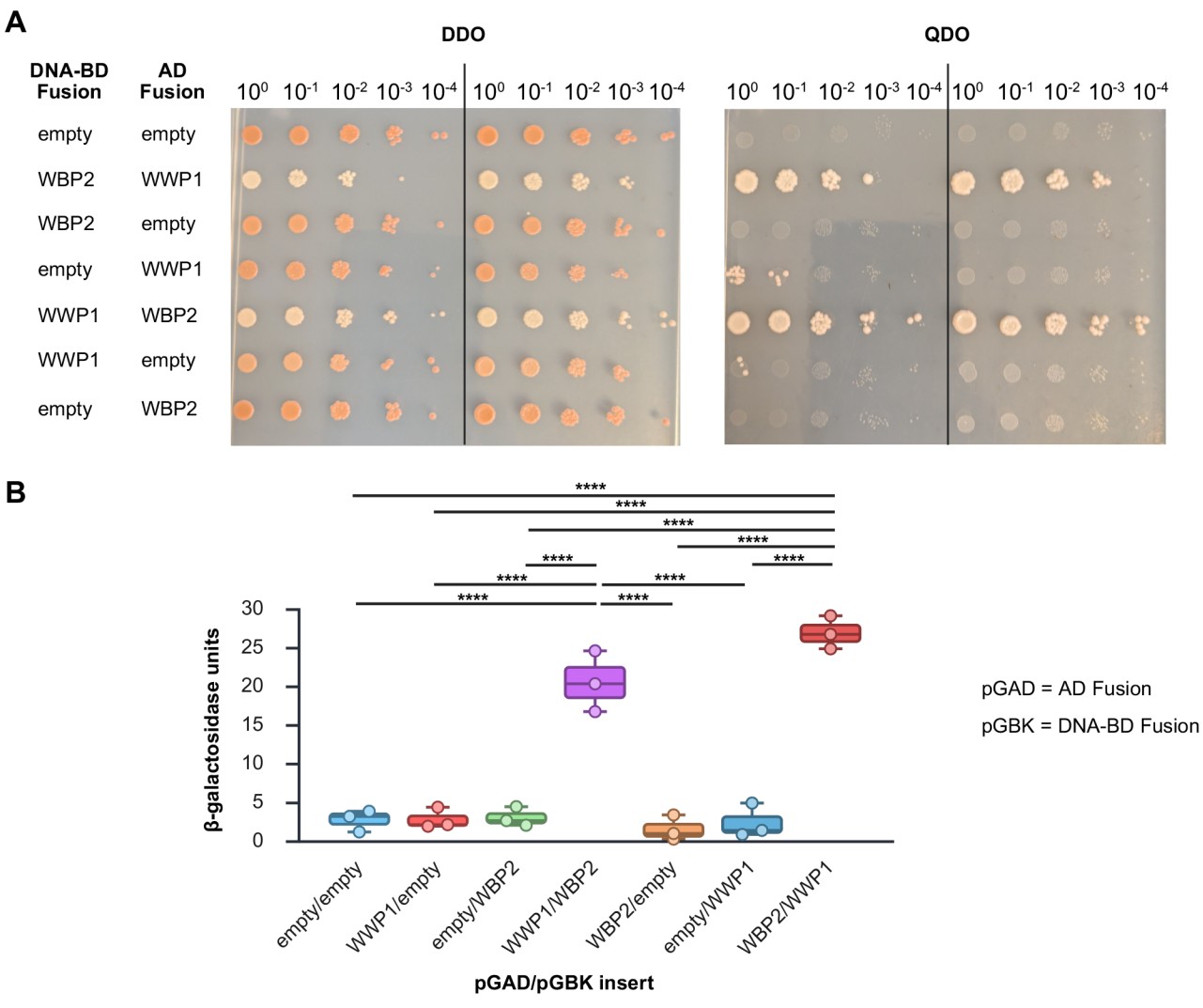

**Fig. 2. Confirmation of the WWP1-WBP2 interaction in the Y2H system.** Two independent co-transformants of the indicated fusion constructs were assessed for activation of the GAL4 reporter genes *ADE2* and *HIS3* on QDO media using a serial dilution assay (A). Irrespective of whether WWP1 or WBP2 was fused with either the GAL4 DNA-BD or with the AD, there was no autoactivation, and growth on QDO plates was seen only when both WWP1 and WBP2 were present in the same cell. This was confirmed by quantifying β-galactosidase activity of the *lacZ* reporter gene of three biological replicates of the same transformation combinations. Data are presented as a box plot with whiskers at minimum and maximum values. Means were compared by one-way ANOVA with Tukey multiple comparisons test. ****$P<0.0001$ (B).

combination (yielding mutants mutPY1+2, mutPY1+3, mutPY2+3, or mutPY1+2+3) using the pGBK-WBP2 construct as a template and then nanopore sequenced to verify that only the desired change was present. Each mutant was co-transformed into haploid AH109 yeast with either pGAD-WWP1 or with pGAD-T7 to assess interaction. While single PY mutants of WBP2 as well as double combinations involving PY1 were still capable of interacting with WWP1 as assessed by serial dilution assay on QDO plates, mutPY2+3 and mutPY1+2+3 were not capable of activating the transcription of the GAL4-dependent reporter genes *HIS3* and *ADE2* when co-transformed with WWP1 (Fig. 4A), similar to negative controls. Interestingly, the quantitative β-galactosidase assay (which measures the expression of the non-essential *lacZ* reporter gene, allowing for more a more sensitive readout of low-levels of interaction and thus of reporter gene expression) revealed that, in combination with WWP1, mutPY2+3 had an intermediate activity between that of the positive control (wild-type pGAD-WWP1 and pGBK-WBP2) and the negative control (pGAD-T7 and

pGBK-T7) whereas the triple PY mutant was statistically indistinguishable from the negative control (Fig. 4B), suggesting that, while the majority of the WWP1-WBP2 binding may occur via the PY2 and PY3 domains, PY1 may also contribute. Importantly, we also showed that all PY mutants were expressed in the yeast system (Fig. 4C).

To determine which WW domains of WWP1 might be interacting with the PY domains of WBP2, a series of in-frame deletions of pGAD-WWP1 were created to delete single WW domain deletions [i.e. ΔWW1 (deletion of amino acid residues 349-381), ΔWW2 (deletion of amino acid residues 382-414), ΔWW3 (deletion of amino acid residues 456-489), or ΔWW4 (deletion of amino acid residues 496-529)] as well as the various double WW domain deletions (ΔWW1+2, ΔWW1+3, ΔWW1+4, ΔWW2+3, ΔWW2+4, and ΔWW3+4). All constructs were sequence verified in their entirety and then co-transformed into haploid AH109 yeast with either pGBK-WBP2 or with pGBK-T7, and interaction was assessed. Using the serial dilution growth assay, it was found that only ΔWW1+3/WBP2

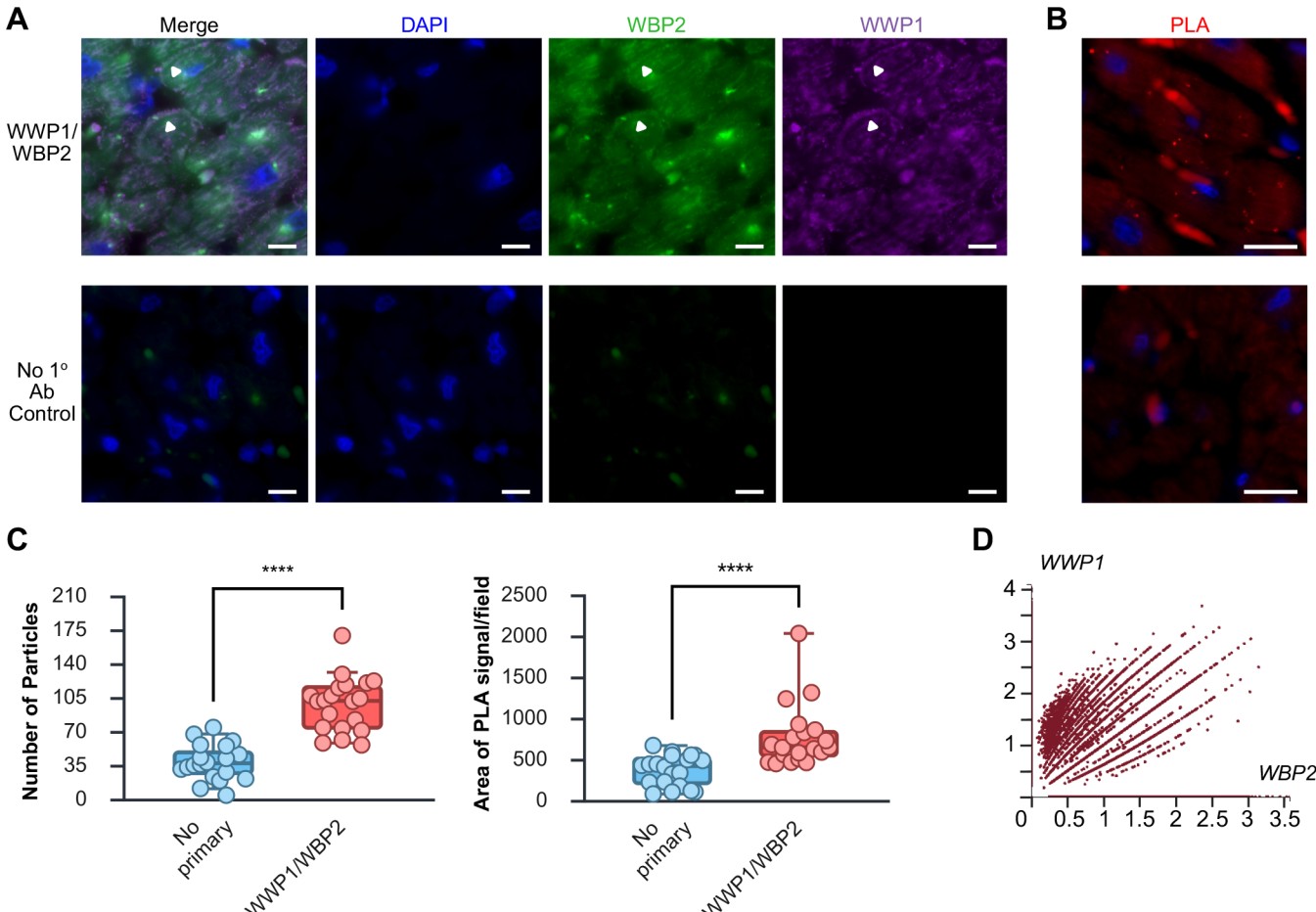

**Fig. 3. Co-localization and interaction of endogenous WWP1 and WBP2 in human heart.** Endogenous co-localization of a subset of cellular WWP1 (purple) and of WBP2 (green) in paraffin sections of human ventricles by co-immunofluorescence (A). Nuclei were counterstained with DAPI (blue). Signal overlap was detected in puncta beneath the sarcolemma (white arrowheads). Scale bars: 10 μm. To determine whether there might be endogenous interaction between the proteins, PLA was performed on two independent sources of cardiac tissue. Representative images, shown in (B), indicate the presence of numerous punctae (red), indicative of proteins localizing within 40 nm of one another, only when both primary antibodies were added. Scale bars: 20 μm. For quantification of PLA signal, at least five random 40x fields were acquired from two different sections of each independent sample and then PLA signal area per field and number of particles per field was determined using ImageJ. Data are presented as a box plot with whiskers representing 5th and 95th percentile and each dot representing data from a 40× field. Mean number of particles was significantly higher in the reactions containing both primary antibodies by two-tailed unpaired $t$-test, while the mean area of PLA signal per field was statistically significantly greater by Mann–Whitney $U$-test ****$P<0.0001$ (C). In a query of the Heart Cell Atlas 2, it was also found that the majority of ventricular myocyte cells/nuclei derived from 25 donors aged 20-75 years co-expressed *WWP1* and *WBP2*. Data are presented as a scatterplot with each dot representing a single cell or nucleus (D).

containing cells were incapable of growth on QDO plates (Fig. 5A), although the growth of the ΔWW2+4/WBP2 co-transformants on QDO was slower than other combinations and did display a more pinkish hue, consistent with a slight reduction in activation of the reporter gene *ADE2*. The dependency upon both WW domains 1 and 3 was confirmed by quantitative β-galactosidase assay (Fig. 5B), which showed that, while all single and double WW domain deletions in combination with WBP2 showed a trend towards decreased β-galactosidase activity as compared to the wild-type WWP1, only the ΔWW1+3/WBP2 combination displayed a statistically significant decrease in reporter gene expression that was comparable to that of the negative controls. Expression of all deletion constructs in yeast was confirmed by western blotting (Fig. 5C). Intriguingly, it appeared that WW domains 1 and 3 showed no preferential binding to either PY2 or PY3 of WBP2 in the Y2H system, for there was no difference in growth on QDO plates by serial dilution assay (Fig. 6A) nor significant alteration in β-galactosidase activity among the ΔWW1/mutPY2, ΔWW1/mutPY3, ΔWW3/mutPY2, and ΔWW3/mutPY3

co-transformants as compared to the WWP1/WBP2 positive controls (Fig. 6B).

## DISCUSSION

Here, using the Y2H system, we have identified 21 putative binding partners for the E3 ubiquitin ligase WWP1 in the heart and verified the endogenous interaction between WWP1 and the oncogenic transcriptional co-activator WBP2. Broadly speaking, previously identified binding partners of WWP1 have been classified as substrates which either contain a PY motif or lack one (Lei et al., 2024; Zhi and Chen, 2012). While the interactions of WWP1 with proteins containing PY motifs are thought to be direct, those without a PY may involve adaptors or larger complexes (Shah and Kumar, 2021). Of those proteins captured in the Y2H screen, 16 out of 21 (76%) have PY motifs (Table 1), which is consistent with the binary nature of PPI typically identified by Y2H analysis (Vidalain et al., 2015). One of the PY-containing interactors, WBP2, has three PY motifs within its C-terminal proline-rich region. These have been

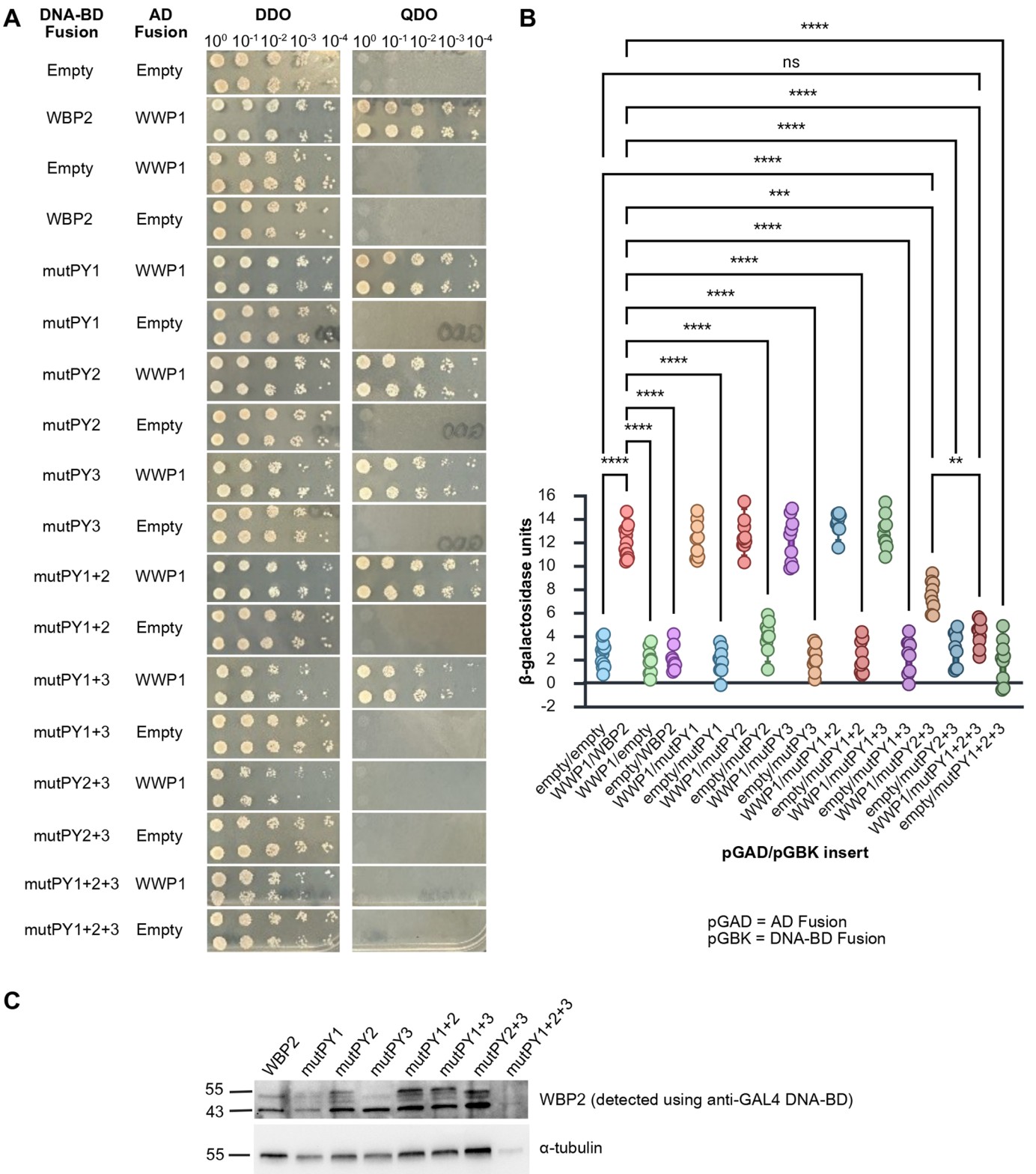

**Fig. 4. PY domains 2 and 3 of WBP2 are required for the interaction with WWP1.** Site directed mutagenesis was employed to alter the three PY domains in WBP2 singly or in combination. These mutants were assessed for their ability to bind to wild-type WWP1 and activate the *ADE2* and *HIS3* reporter genes in the Y2H system using a serial dilution assay (A). WBP2 with mutated PY domains 2 and 3 (mutPY2+3) or the triply mutant WBP2 (mutPY1+2+3) were unable to support growth on QDO media when co-transformed with WWP1. The degree to which the WWP1-WBP2 interaction was impaired was tested using a quantitative β-galactosidase assay (B). Data are presented as a box plot with whiskers representing 5th and 95th percentile and each dot representing an independent biological replicate. Means were compared by Welch's one-way ANOVA with Games-Howell multiple comparisons test. mutPY2+3 displayed an activity intermediate that of the wild-type proteins and mutPY1+2+3. Expression of all mutant constructs was confirmed by western blotting using an antibody that recognizes the DNA-binding domain of GAL4 (C). ns=not significant, **$P<0.01$, ***$P<0.001$, ****$P<0.0001$.

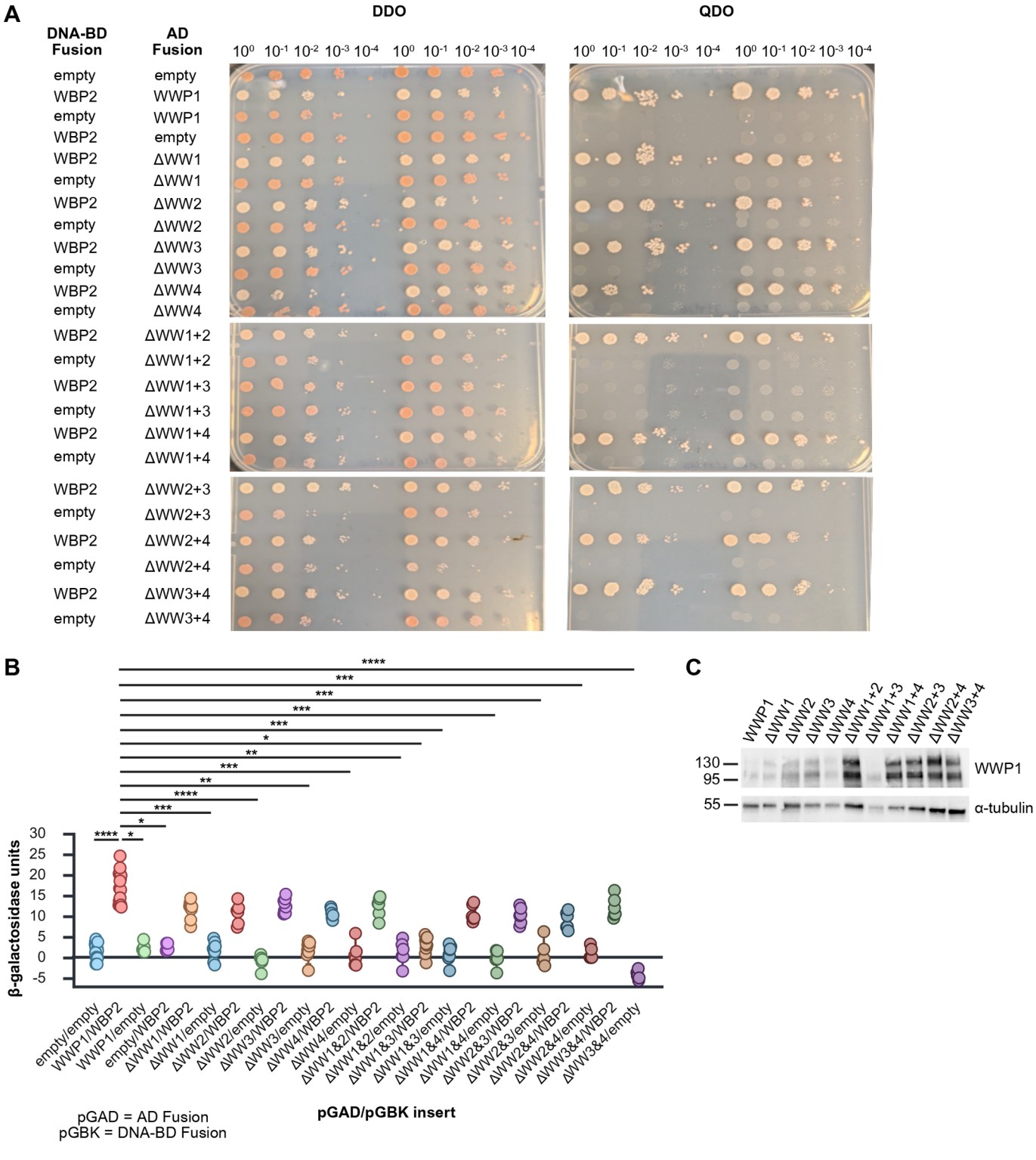

**Fig. 5. WW domains 1 and 3 of WWP1 are required for the interaction with WBP2.** Site directed mutagenesis was employed to precisely delete the four WW domains of WWP1 singly or in combination. These mutants were assessed for their ability to bind to wild-type WBP2 and activate the *ADE2* and *HIS3* reporter genes in the Y2H system using a serial dilution assay (A). Only WWP1 lacking both WW domains 1 and 3 was unable to interact with WBP2 and thus could not grow on QDO plates. The degree to which the WWP1-WBP2 interaction was impaired was tested using a quantitative β-galactosidase assay (B). Data are presented as a box plot with whiskers representing 5th and 95th percentile and each dot representing an independent biological replicate. Means were compared by Kruskal–Wallis test with Dunn's multiple comparisons test. Yeast co-transformed with ΔWW1+3 and WBP2 showed β-galactosidase activity that was statistically indistinguishable from that of any of the negative controls. Expression of WWP1 deletion constructs was detectable in total yeast protein extracts by western blot (C). *$P<0.05$, **$P<0.01$, ***$P<0.001$, ****$P<0.0001$.

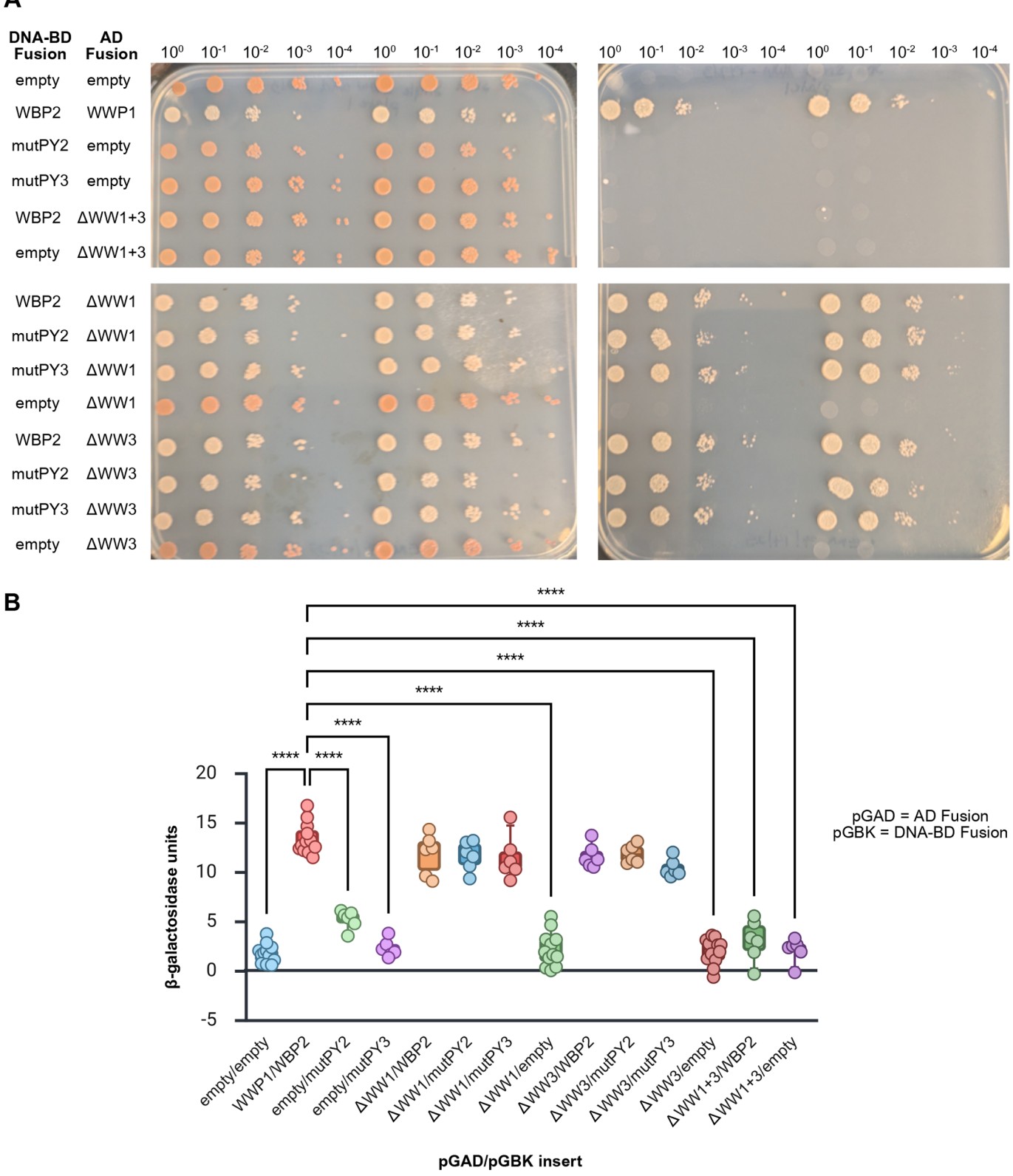

**Fig. 6. PY2 and PY3 of WBP2 can bind to either WW1 or WW3 of WWP1.** To determine whether WW domains 1 and 3 showed a preference for either PY2 or PY3 of WBP2, various combinations of WWP1 and WBP2 mutants were co-transformed into yeast to evaluate their ability to promote the expression of the *ADE2* and *HIS3* reporter genes using a serial dilution assay (A). Single WW mutants paired with either mutPY2 or mutPY3 could grow on media lacking adenine and histidine, indicating both binding configurations were possible in this system. This was confirmed using quantitative β-galactosidase assays (B). Data are presented as a box plot with whiskers representing 5th and 95th percentile and each dot representing an independent biological replicate. Means were compared by Welch's one-way ANOVA with Games-Howell multiple comparisons test. ***$P<0.001$, ****$P<0.0001$.

shown to mediate its association with a number of different WW domain-containing proteins. Specifically, PY2 mediates WBP2's association with the transcriptional co-activator TAZ (Chan et al., 2011) and the E3 ubiquitin ligase NEDD4 (Jolliffe et al., 2000), while PY3 is involved in binding to the transcriptional co-activator YAP (McDonald et al., 2011) and the tumor suppressor WWOX (McDonald et al., 2012). In contrast, all three PY domains are required for the interaction between WBP2 and the scaffolding protein WWC3 (Han et al., 2021).

Interestingly, both PY2 and PY3 of WBP2 facilitate its interaction with the WWP1 paralog ITCH (Lim et al., 2016), similar to what we observed here. Further, WW domains 1 and 3 from both ITCH and from WWP1 were necessary for interacting with WBP2, suggesting that these ligases might compete for access to this binding partner *in vivo*. In breast cancer, ITCH has been described as a negative regulator of WBP2, promoting its ubiquitination and proteasomal degradation (Lim et al., 2016). In contrast, WBP2 is reported to be upregulated in over 80% of breast cancer patients and is associated with poorer overall survival and disease-free survival outcomes (Chen et al., 2007b; Lim et al., 2016, 2011; Song et al., 2018). Likewise, genomic amplification of *WWP1* (Chen et al., 2007a) as well as high protein level expression of WWP1 has been noted in breast tumors (Chen et al., 2007a, 2009; Jiang et al., 2024; Nguyen Huu et al., 2008) and has additionally been reported in luminal breast cancer that metastasize to the bone (Jiang et al., 2024). The positive correlation between WWP1 and WBP2 protein expression in breast cancer initiation and progression implies that, unlike the interaction of ITCH and WBP2, the WWP1–WBP2 interaction might not lead to the degradation of WBP2. Rather, these observations offer the tantalizing possibility that WBP2 may act as an adaptor protein for WWP1 to relieve the allosteric autoinhibition mediated by the interaction among WW2, WW4, and the linker region between WW2 and WW3 and the HECT domain (Wang et al., 2019).

In mammals, an increasing number of proteins, such as NDFIP1, NDFIP2, α- and β-arrestins, or ENTREP have been shown to serve as activators or adaptors for multiple members of the NEDD4 family of E3 ubiquitin ligases (Shah and Kumar, 2021; Tsunoda et al., 2022). Due to the differing structure and subcellular localization of these adaptors, it has been hypothesized that the E3-adaptor pairing may vary with cell type and under physiological or pathological conditions to influence E3 localization and catalytic activity as well as to fine-tune E3-substrate interactions. Like the other NEDD4 family adaptors, WBP2 can interact with multiple members of the family, including NEDD4, NEDD4L, and the WWP1 paralogs ITCH and WWP2 (Ingham et al., 2005; Jiang et al., 2019; O'Connor et al., 2015; Persaud et al., 2009; Pirozzi et al., 1997). Further, it has been reported that WBP2 can translocate from the cytoplasm to the nucleus when tyrosine phosphorylated downstream of EGFR and WNT signaling in MB231 and HeLa cells (Lim et al., 2011). While loss of WBP2 function has been implicated in progressive high-frequency hearing loss in mouse and humans (Buniello et al., 2016), little is known about potential physiological functions of WBP2 in other tissue types. In this report, we show that low levels of endogenous WBP2 can be detected in human ventricular cardiomyocytes, most notably in puncta near the sarcolemma where WWP1 is also present (Fig. 3A). Sarcolemmal WWP1 has also been described in skeletal muscle fibers of chickens (Imamura et al., 2016). Although there was some endogenous level of interaction of WWP1 and WBP2 in normal ventricular cardiomyocytes (Fig. 3B,C), these complexes might increase with stress or injury. Interestingly, in cardiac muscle, there

is a discrete population of mitochondria that resides in this space which functions in signaling, in ion homeostasis, and in regulation of cell membrane potential. As such, these subsarcolemmal mitochondria may be more susceptible to damage under certain stress conditions (Hollander et al., 2014). This is particularly remarkable in light of the fact that three mitochondrial proteins were also identified as WWP1 interactors in our Y2H screen. Thus, the WWP1 interaction network might be spatially positioned in the heart to respond to stress and its dysfunction could substantially contribute to cardiac aging.

There are limitations to the current study. Although the Y2H screen described in this report was conducted with high stringency screening and eliminated satellite prey plasmid contaminants through serial single colony purification, there is still a possibility of false positives. The only interaction that was confirmed to endogenously occur in the heart was that of WWP1 and WBP2 by PLA in a limited number of clinical samples. Because of the low level of expression of WBP2 as well as the antibody quality, endogenous co-IP was not attempted. Since the Y2H system depends on direct PPI, we also might have missed larger complexes of which WWP1 is a part, though the possibility exists that other potential adaptors (like TXNIP) were captured in this screen. Further, because cardiomyocytes occupy the majority of the heart's volume, their expressed genes will be more highly represented in the library than will be those of non-myocytes so WWP1-mediated interactions in non-myocytes could have been obscured by this process. However, the Y2H system was ideal for biochemically dissecting the respective domains of WWP1 and WBP2 that mediate the interaction between them. While future studies are needed to determine whether or not the binding of WWP1 to WBP2 promotes degradative ubiquitination of WBP2 and what the functional relevance of this interaction might be in various cellular and disease contexts, the findings reported here suggest that therapeutic strategies aimed at reducing WWP1 activity through administration of inhibitors like indole-3-carbinol (Fan et al., 2024; Lee et al., 2019; Lu et al., 2023; Pennarossa et al., 2023; Sax et al., 2024) might need to account for how putative allosteric adaptors like WBP2 could factor into those approaches. In total, these findings represent the critical first step in defining the WWP1 interactome in order to devise strategies to maintain cellular proteostasis with age.

## MATERIALS AND METHODS
### Plasmid construction
The full-length ORF of human WWP1 (2769 bp) was amplified from pcDNA3-MYC-WWP1 (generously provided by Dr. Yu-Ru Lee) using Phusion High-Fidelity DNA polymerase (Thermo Fisher Scientific) and directionally cloned in frame into the EcoRI/BamHI sites of both the pGBK-T7 'bait' and pGAD-T7 'prey' yeast expression vectors for use in the GAL4-based Y2H system (Takara). Similarly, the full length WBP2 open reading frame (786 bp) was amplified from one of the clones captured in the Y2H screen using Phusion High-Fidelity DNA polymerase (Thermo Fisher Scientific) and directionally cloned in frame into the EcoRI/BamHI sites of both the pGAD-T7 and of pGBK-T7 plasmids. All PY mutants of pGBK-WBP2 and WW deletions of pGAD-WWP1 were generated using the Q5 Site-Directed Mutagenesis kit (New England Biolabs). Primer sequences used for amplification and site-directed mutagenesis are listed in Table 2. All constructs were nanopore sequenced in their entirety (Plasmidsaurus) before use.

### Y2H screen
The pGBK-WWP1 'bait' plasmid was transformed into haploid AH109 [*MATa, trp1-901, leu2-3, 112, ura3-52, his3-200, gal4Δ, gal80Δ, LYS2:: GAL1_{UAS}-GAL1_{TATA}-HIS3, GAL2_{UAS}-GAL2_{TATA}-ADE2, URA3::MEL1_{UAS}-MEL1_{TATA}-lacZ*, (James et al., 1996)] cells using the Frozen-EZ yeast

**Table 2. List of primers used in plasmid construction**

| Primer name | Primer sequence |
| --- | --- |
| EcoRI hWWP1 F | 5′-TAGTAAGAATTCATGGCCACTGCTTCACCAAG-3′ |
| BglIII hWWP1 R | 5′-CGAAGATCTTCATTCTTGTCCAAATCCCTCTGT-3′ |
| WBP2 EcoRI F | 5′-CAGTAGGAATTCATGGCGCTCAACAAGAAT-3′ |
| WBP2 BamHI R | 5′-CTCGGATCCCTACTGGGTCTTCTTATCTTC-3′ |
| WBP2 PY1 P168A F | 5′-CCCCTGCCCT**gct**GGCTACCCCT-3′ |
| WBP2 PY1 P168A R | 5′-TACATTCCATTGGCGACTGGCGGG-3′ |
| WBP2 PY2 P198A F | 5′-GCCCCCACCA**gcg**CCCTACCCTG-3′ |
| WBP2 PY2 P198A R | 5′-TGCACGTATCCCATGGCCCCG-3′ |
| WBP2 PY3 P250A F | 5′-GCCGCCGCCA**gct**CCCTACTACC-3′ |
| WBP2 PY3 P250A R | 5′-TGGCTCGTGGGCATGTAGACGTTG-3′ |
| WWP1 ΔWW1 F | 5′-CAACCTTTACCTCCAGGTTG-3′ |
| WWP1 ΔWW1 R | 5′-TGTGTTGGCATTCCCAGAC-3′ |
| WWP1 ΔWW2 F | 5′-GAATCTGTCCGAAATTTTGAACAG-3′ |
| WWP1 ΔWW2 R | 5′-AGGTTGTGGTCTCTCCCATG-3′ |
| WWP1 ΔWW3 F | 5′-CAAGGCTTACAGAATGAAG-3′ |
| WWP1 ΔWW3 R | 5′-ATAAGGGTCATTTTCTGC-3′ |
| WWP1 ΔWW4 F | 5′-GGGAAGTCATCTGTAAC-3′ |
| WWP1 ΔWW4 R | 5′-TTCATTCTGTAAGCCTTG-3′ |

Amino acid substitution indicated in lower-case, bold font.

transformation II kit (Zymo) according to the manufacturer's instructions. Transformants were selected on synthetic defined media lacking tryptophan (SD-Trp) and assessed for autoactivation by looking for expression of the *MEL1*, *HIS3*, and *ADE2* reporter genes in the absence of any 'prey' construct. To identify WWP1 interactors, the AH109 transformant strain was then mated to a pre-transformed Mate & Plate *Homo sapiens* cDNA heart library (Takara) which contained $1.15 \times 10^7$ independent clones with an insert size range of 0.5 to >3 kb in the 'prey' pGAD vector, which contains the GAL4 activation domain. A total of $3.5 \times 10^7$ diploids were screened (cfu/mL of diploids x resuspension volume). Interactors were selected for on synthetic defined media lacking leucine, tryptophan, histidine, and adenine (quadruple drop out or QDO) and then single colony purified three times.

'Prey' plasmids were subsequently isolated from positive diploids using the QIAprep Spin Miniprep Kit (Qiagen) with slight modification. Briefly, single colonies were grown in synthetic defined medium lacking leucine and tryptophan (double drop out or DDO) liquid medium for 16–24 h at 30°C with shaking (250 rpm). Cultures were pelleted (5 min at $5000 \times g$ in a microfuge) and resuspended in Buffer P1. 50-100 µl of acid-washed glass beads (425-600 µm, Sigma) were added to each tube and vigorously vortexed for 5 min. The remainder of the plasmid isolation was conducted according to the manufacturer's instructions. Plasmids were assessed for cDNA inserts by PCR using primers flanking the cloning site (5′-CTATTCGATGATGAAGA-TACCCCACCAAACCC −3′) and (5′-GTGAACTTGCGGGGTTTTTCAG-TATCTACGAT-3′). Those containing detectable inserts were transformed into chemicompetent DH5α cells and propagated using the pGAD-T7 ampicillin resistance marker. Duplicate clones were identified by Sau3AI restriction enzyme (New England Biolabs) digest analysis. Unique clones were sequenced from the T7 promoter. Sequence information was used for BLAST searches in GenBank to reveal the identity of the 'prey' construct insert.

### Serial dilution assay

For each 'bait'/'prey' pairwise comparison analyzed, two independent single colonies of co-transformed haploid AH109 yeast that were cultured on DDO plates for no more than 2 weeks were picked and grown in 3 ml of DDO liquid media in a shaking incubator at 30°C (250 rpm) until saturated. The concentration of each culture was adjusted to 1 $OD_{600}$ per ml with DDO media. A serial dilution series ($10^0$, $10^{-1}$, $10^{-2}$, $10^{-3}$, $10^{-4}$) was prepared for

each sample. Five microliters of each dilution were spotted onto DDO and QDO plates under sterile conditions, allowed to dry, and then incubated in a 30°C incubator for 3-4 days. Growth was documented using a standard cell phone camera.

### Quantitative beta-galactosidase assay

To determine the strength of interaction between a 'bait' and 'prey' construct in co-transformed haploid AH109 yeast, quantitative β-galactosidase assays were conducted as previously described (Trimborn et al., 2022) with slight modification. Briefly, three biological replicates were prepared for each sample by picking five single colonies each from fresh DDO plates into sterile ddH$_2$O. Cells were vortexed and then spotted (ten spots per biological replicate) onto a separate DDO plate for each biological replicate to avoid any plate-specific effects. After a 48 h incubation at 30°C, cells from all spots of a given biological replicate were harvested into 1 ml of Z-buffer (60 mM Na$_2$HPO$_4$•2H$_2$O, 40 mM NaH$_2$PO$_4$•H$_2$O, 10 mM KCl, 1 mM MgSO$_4$•7H$_2$O, pH=7.0) and the $OD_{600}$ measurement was determined and standardized to 5 for each sample. Cells were permeabilized via three freeze-thaw cycles in liquid nitrogen and a 30°C water bath, respectively, and resuspended in a final volume of 100 µl of Z-buffer. Two technical replicates of equal volume for each biological replicate were arrayed into a white 96-well plate (Greiner bio-one, cat. no. 655083). Blanks were created with equal volumes of Z-buffer. To each well containing cellular extracts or buffer alone, 330 µl of X-gal staining solution (0.3% agarose, 0.3% 5-Bromo-4-chloro-3-indolyl β-D-galactopyranoside, and 0.06% β-mercaptoethanol in phosphate buffered saline, PBS, pH=7.4) was added and gently pipetted to mix thoroughly. Color development was documented initially (t$_0$) and in 30 min increments over the course of 6 h using a cell phone camera. β-galactosidase activity was determined using the ReadPlate3.0 plugin for ImageJ after subtracting both the average blank $A_{uncorr}$ and initial (t$_0$) $A_{uncorr}$ value from each well (to account for the pink color of Ade2 deficient cells, which, when converted to grayscale, was incorrectly scored as β-galactosidase activity). The timepoint used for analysis of each plate was determined by presence of robust blue precipitate in the positive control with no blue color appearing in negative controls and establishment of being within the linear range. For the WWP1-WBP2 interaction, this was at t$_{90}$ or t$_{120}$. All assays were conducted using the same lot of X-gal (Combi-Blocks, SS-7694, Batch D14745).

### Immunofluorescence

Paraffin sections (5 µm in thickness) of human ventricular tissue derived from two independent sources (Zyagen and BioChain) were deparaffinized in xylenes and decreasing concentrations of ethanol before being rehydrated in PBS (pH=7.4). Endogenous peroxidases were inactivated in a solution of 3% hydrogen peroxide and 100% methanol combined in a 1:1 ratio. Antigen retrieval was subsequently performed in boiling 10 mM citric acid (pH=6.0) buffer. To reduce non-specific binding of the primary antibodies, the sections were incubated in 1% Tyramide Signal Amplification (TSA) block (Life Technologies) before primary antibody application. A cocktail containing a 1:100 dilution of a mouse anti-human WWP1 monoclonal antibody (clone 1A7, Abnova #H00011059-M01, RRID:AB_509107) and a 1:200 dilution of a rabbit anti-human WBP2 polyclonal antibody (Atlas Antibodies #HPA065682, RRID:AB_2685530) was applied to one of the sections on the slide. The other section received only 1% TSA block and served as a no primary negative control. Incubation was performed overnight at 4°C. To bind and detect the WBP2 primary antibody, a donkey anti-rabbit biotin-conjugated secondary antibody (1:200 dilution, Life Technologies A1603) followed by a streptavidin-conjugated horseradish peroxidase (1:100 dilution, Thermo Fisher Scientific) and then TSA with AlexaFluor594 was utilized whereas the WWP1 primary was directly detected with a 1:500 dilution of donkey anti-mouse secondary antibody conjugated to AlexaFluor647 (Jackson ImmunoResearch). Nuclei were counterstained with 4′,6-diamidino-2-phenylindole (DAPI) before coverslips were mounted with ProLong Gold Antifade (Thermo Fisher Scientific). After curing, sections were imaged using a Zeiss AxioVision A1 microscope with Axiocam MRm monochromatic digital camera. Images were pseudo-colored to maximize contrast and overlap of signals. Exposure times as well as brightness and contrast thresholds were set for the

experimental samples and then applied to the no primary antibody negative control. At least two sections from each source were analyzed for each condition in order to obtain representative images.

## PLA

PLA experimental and negative controls (where one or both primary antibodies were left out of the reaction) were performed on at least two sections of heart tissue derived from two independent donors (Zyagen and BioChain) according to the method described by Hegazy et al. (2020) using the Duolink In Situ Orange starter kit for mouse/rabbit (Millipore Sigma, #DUO92102). Primary antibodies used were mouse anti-human WWP1 (clone 1A7, Abnova #H00011059-M01, RRID:AB_509107, 1:100) and rabbit anti-human WBP2 (Atlas Antibodies #HPA065682, RRID: AB_2685530, 1:50). Antigen retrieval was performed in Tris-EDTA Antigen Retrieval Buffer (pH 9.0, Proteintech) for 15 min at 95°C. At least five 40x fields were randomly collected from each section using a Zeiss AxioVision A1 microscope with Axiocam MRm monochromatic digital camera. Acquired images were split into individual channels and then converted to 8-bit grayscale in ImageJ (version 1.54p, http://imagej.org). All were auto thresholded using the Triangle setting. PLA signal area was then measured for the entire section. Afterwards, the section was masked, some background removed (using Process>Binary>Open), and coalescing signals were segmented using the watershed feature. Individual particle counts for each section were enumerated using the Analyze Particles feature of ImageJ.

## Western blotting

Total yeast proteins were extracted from 50 ml of a mid-log phase culture of yeast grown in DDO medium utilizing mechanical lysis with acid-washed glass beads (425-600 μm, Sigma) in extraction buffer (10 mM $NaPO_4$, 5 mM EDTA, 50 mM NaCl, 0.1% Triton X-100, pH 7.8) supplemented with protease inhibitors. Protein concentrations were quantified using the BCA Protein Assay Kit (Pierce) as directed by the manufacturer. Twenty-five micrograms of protein was heat-denatured before being separated on a 4-15% tris-glycine gradient gel (MP Biochemicals) and transferred to a PVDF membrane. Membranes were blocked in 5% (w/v) non-fat dry milk in PBST (PBS containing 0.05% Tween-20) for 1 h at room temperature with gentle agitation. Membranes were then incubated at 4°C overnight in primary antibodies (mouse anti-yeast GAL 4 DNA-binding domain, clone RK5C1, Santa Cruz Biotechnology #sc-510, RRID:AB_627655, 1:200; mouse anti-human WWP1, Abnova #H00011059-M01, RRID: AB_509107, 1:500; rabbit anti-yeast α-tubulin, clone EPR13799, Abcam #ab184970, RRID:AB_2928998,1:15,000) diluted in blocking buffer. Species specific HRP-conjugated secondary antibodies (sheep anti-mouse IgG #NA931, GE Healthcare or goat anti-rabbit IgG #7074S, Cell Signaling Technology) and SuperSignal West Femto Maximum Signaling Substrate (Thermo Fisher Scientific) were used for signal detection and membranes were imaged using a Bio-Rad ChemiDoc Touch gel imaging system. Membranes were first blotted for either the GAL4 DNA-BD or for WWP1 before being stripped in boiling 5 mm EDTA (pH 8.0) and reprobed with the anti-α-tubulin antibody as a loading control.

## Statistics

To compare sample means across multiple groups with one another within a single experiment (where the variances of the samples were homogenous by Levene test for equality of variances and the distributions of the samples were normal by the Shapiro-Wilk normality test), one-way ANOVA with Tukey multiple comparisons test was run using R 4.2.2 in BioRender Graph. Means from samples pooled across multiple experiments were compared by Welch's one-way ANOVA with Games-Howell multiple comparisons test (where the variances of the samples were not homogeneous but the distributions of the samples were normal by Shapiro-Wilk) or by the nonparametric Kruskal–Wallis test with Dunn's multiple comparisons if non-normal distributions were detected using a Shapiro-Wilk normality test (R 4.2.2 in BioRender Graph). To compare the means of two groups with normal distributions (by Shapiro-Wilk) and equal variances (by Levene test of equal variance), a two-tailed, unpaired $t$-test was used, whereas if the distributions were not normal, the means of the two groups were compared by the nonparametric Mann–Whitney $U$-test (using R 4.2.2 in BioRender Graph). For all measures, an alpha-value <0.05 was considered statistically significant.

## Acknowledgements
The authors gratefully acknowledge Carlos Alfaro Quinde for assistance with the molecular cloning of the pGBK-WWP1 bait vector, as well as Dr. Michael D. Wyatt and Jordan Headen for helpful discussions on western blotting and particle analysis for PLA, respectively.

## Competing interests
The authors declare no competing or financial interests.

## Author contributions
Conceptualization: D.L.P., L.E.M.; Formal analysis: M.E.A., D.L.P., L.E.M.; Funding acquisition: M.E.A., L.E.M.; Investigation: M.E.A., Y.W., N.K.G., L.E.M.; Methodology: M.E.A., N.K.G., D.L.P., L.E.M.; Project administration: L.E.M.; Resources: D.L.P., L.E.M.; Supervision: L.E.M.; Validation: M.E.A., Y.W., N.K.G., D.L.P., L.E.M.; Visualization: M.E.A., D.L.P., L.E.M.; Writing – original draft: M.E.A., L.E.M.; Writing – review & editing: Y.W., N.K.G., D.L.P., L.E.M.

## Funding
This work supported in part by an undergraduate research grant from the University of South Carolina Honors College (M.E.A.), an undergraduate research enhancement program grant from the University of South Carolina McCausland College of Arts and Sciences (M.E.A.), a Magellan mini-grant from the University of South Carolina Vice President for Research (M.E.A.), and a mid-career faculty developmental fellowship from the University of South Carolina McCausland College of Arts and Sciences (L.E.M.). Open Access funding is provided by a Read & Publish agreement with the University of South Carolina. Deposited in PMC for immediate release.

## Data and resource availability
All relevant data and details of resources can be found within the article and its supplementary information.

## Peer review history
The peer review history is available online at https://journals.biologists.com/bio/lookup/doi/10.1242/bio.062347.reviewer-comments.pdf

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
