## [Peer Review File · Biology Open]

Identification and Validation of an Interaction between the E3 Ubiquitin Ligase WWP1 and the Transcriptional Co-Activator WBP2 in the Human Heart

Meaghan E. Arnold, Ymani Wright, Natalie K. Grantham, Douglas L. Pittman and Lydia E. Matesic

DOI: 10.1242/bio.062347

Editor: Luca Scorrano

Review timeline

Original submission:	29 October 2025
Editorial decision:	3 November 2025
First revision received:	2 March 2026
Accepted:	18 March 2026

Original submission

First decision letter

MS ID#: bio.062347

MS Title: Identification and Validation of an Interaction between the E3 Ubiquitin Ligase WWP1 and the Transcriptional Co-Activator WBP2 in the Human Heart

Authors: Meaghan E. Arnold; Ymani Wright; Douglas L. Pittman; Lydia E Matesic

I have now reached a decision on the above manuscript.

The reviewer reports are shown at the bottom of this email.

As you will see, the reviewers raised a number of substantial criticisms that prevent me from accepting the paper at this stage.

They suggest, however, that a revised version might prove acceptable, if you can address their concerns. If you think that you can deal satisfactorily with the criticisms on revision, I would be pleased to see a revised manuscript. We would then return it to the reviewers.

At this stage, we also ask you to ensure your manuscript complies with our formatting guidelines. Provided you are able to fully address the referees' comments, we are positive about publication of your paper (we accept over 95% of revision submissions) and therefore hope you won't mind any extra work involved in reformatting your manuscript at this point.

Please upload both a 'clean' version of your Word file, along with a highlighted version clearly showing where you have made changes in the revised manuscript. Please avoid using 'Track changes' in Word files as these are lost in PDF conversion.

I should be grateful if you would also provide a point-by-point response detailing how you have dealt with the points raised by the reviewers in the 'Response to Reviewers' box. Please attend to all of the reviewers' comments. If you do not agree with any of their criticisms or suggestions please explain clearly why this is so.

Reviewer 1

Comments for the author

Arnold et al. are interested in the WW-domain containing protein 1 (WWP1), an E3 ubiquitin ligase which deregulation is associated with several diseases. The authors set out to obtain a comprehensive view of the proteins that interact with WWP1, particularly with protein expressed in heart tissues. They perform a yeast two-hybrid screen with a collection of genes expressed in human heart cells and WWP1 as a bait. 21 hits are identified in this screen, with 11 being identified several times, suggesting that the library was well covered during the screen. The WWP1-WBP2 interaction, already known from multiple experimental approaches, is further explored. Beta-galactosidase assays or swapping the bait and prey confirm that WWP1 and WBP2 interact in the yeast two-hybrid system. Using antibodies to WWP1 and WBP2, the authors stained human ventricular tissues. They argue that the two signals do overlap to some extent.

They then mutated the 3 PY domains in WBP2 (second proline to alanine). The yeast two-hybrid experiments suggest that PY domains 2 and 3 may mediate most of the interaction between WWP1 and WBP2. Finally, they mutated the WW domains of WWP1 and found that WW domains 1 and 3 are required for the interaction with WBP2. They do not find any difference in binding to the different PY domains.

The yeast two-hybrid data is quite clear and present new protein-protein interactions for WWP1, which will be valuable for the field. However, I have a few suggestions to improve the strengths of the conclusions proposed by the authors.

1. Concerning the microscopy data (Fig2C): There is no quantification/analysis of the images beyond showing on the images two occurrences of signal overlap. My understanding from the text is that the immunofluorescence was performed only once. The no-primary antibody control is rather nice, although there is some staining for the WBP2 channel but I find it very difficult to reach any firm conclusion from the images presented. It would require more work to conclude that the two proteins do indeed localise to the same sub-cellular structures, yet even if they do co-localise, it does not mean that they necessarily interact *in vivo*. I therefore suggest to remove this statement from the text (for example, line 247: "The only interaction that was confirmed to endogenously occur in the heart was that of WWP1 and WBP2 by co-immunofluorescence."), to quantify the extent of co-localisation and to clarify whether the immunofluorescence staining was performed once or more.

2. The yeast two-hybrid system is used to quantify the interaction between several mutants of WWP1 and WBP2. The data appears very clear but, in my view, lacks a major control. The protein level of each of the mutant and wild-type proteins should be quantified using western-blot. If the mutants that show a reduced beta-galactosidase activity are present at a comparable level to that of the wild-type protein, then it will be possible to conclude on the strength of the interactions. This is true for the WWP1 and WBP2 mutants.

Minor typo in the abstract: proteosome instead of proteasome.

Reviewer 2

Comments for the author

The study from Arnold M. and colleagues investigates the interaction between the E3 ubiquitin ligase WWP1 and the transcriptional co-activator WBP2 in the human heart. A previous study by the authors described the role of WWP1 in a mouse model. Here, using a yeast two-hybrid (Y2H) screen using a human heart library, the authors identified 21 potential WWP1 interactors and selected WBP2 for further experimental validation. The WWP1-WBP2 interaction was confirmed through additional Y2H assays using either full-length proteins or mutated versions lacking specific PY motifs or WW domains. Based on these results, the authors conclude that the PY2 and PY3 motifs of WBP2 can bind to either the WW1 or WW3 domains of WWP1. Finally, they tested whether this

interaction occurs in the human heart using fluorescence microscopy to assess co-localization, observing partial overlap of both proteins.

Overall, this is a well-written and carefully executed study that provides a valuable list of WWP1 interactors and characterizes one of these interactions through canonical PY-WW motif binding. However, the major claim that the WWP1-WBP2 interaction occurs in the human heart is not sufficiently supported by the presented data. The authors should revise these statements throughout the manuscript or provide additional experimental evidence supporting the claim.

Major Issues

1. Unsupported claim of interaction in the human heart

The manuscript repeatedly claims that the interaction between WWP1 and WBP2 occurs in the human heart. This is a major conclusion that is not supported by the current results. The only evidence presented is a widefield immunofluorescence microscopy experiment, which lacks the spatial resolution necessary to demonstrate protein-protein interactions (widefield fluorescence microscopy is ~250nm). At this stage, the data only show that WWP1 and WBP2 may be present in the same regions or compartments, not that they physically interact. Moreover, the staining patterns suggest that most of the signal is not overlapping.

To support this conclusion, the authors should modify the claim to state that the two proteins co-localize (not necessarily interact) or perform additional experiments, such as co-immunoprecipitation or proximity ligation assays, using human heart tissue to demonstrate a physical interaction.

Specific instances where the claim should be revised include:

* Line 247: "The only interaction that was confirmed to endogenously occur in the heart was that of WWP1 and WBP2 by co-immunofluorescence."

* Line 112: "Demonstrate that this interaction can occur endogenously in the heart."

Minor Issues

1. The meaning of the phrase "Chief of the interactors" is unclear, since WBP2 ranks as the fourth hit by the number of colonies in the Y2H screen, with MTCO1 being the top interactor. The authors should explain why WBP2 was chosen for further study despite not being the top hit (e.g., biological relevance, previous literature, or domain complementarity).

2. The color coding in Figure 1 uses non-standard p-value thresholds. The authors should clarify why they did not use conventional cutoffs (e.g., 0.01 or 0.001).

3. The PY mutants should be labeled in a consistent format with the WW mutants, e.g., Δ PY1, Δ PY2-3, etc.

4. The authors should explain why the Δ PY2-3 double mutant shows different results in the QDO medium versus the β -galactosidase assays.

5. Figure 5 could include the pairings of the individual WW mutants with the double PY mutants, particularly Δ PY2-3, which showed partial loss of interaction in the β -galactosidase assay. Since the authors already have the tools it is a simple experiment to do.

Reviewer's Responses to Questions

Experimental quality

Does each figure have the proper controls?

If 'No', please indicate reasons in Comments for Author box below.

Reviewer #1:

- No

Reviewer #2:

- Yes

Were the data analyzed using appropriate statistical tests?

If 'No', please indicate reasons in Comments for Author box below.

Reviewer #1:

- Yes

Reviewer #2:

- Yes

Reproducibility

Were experiments performed using adequate number of biological replicates?

If 'No', please indicate reasons in Comments for Author box below.

Reviewer #1:

- No

Reviewer #2:

- Yes

Does the methods section provide sufficient detail to permit reproducibility?

If 'No', please indicate reasons in Comments for Author box below.

Reviewer #1:

- Yes

Reviewer #2:

- Yes

Completeness

Are the manuscript's conclusions supported by the data?

If 'No', please indicate reasons in Comments for Author box below.

Reviewer #1:

- No

Reviewer #2:

- No

Scholarship

Do the authors cite and discuss the merits of data that would argue for and against their conclusion?

If 'No', please indicate reasons in Comments for Author box below.

Reviewer #1:

- Yes

Reviewer #2:

- Yes

Does the manuscript title & abstract accurately reflect the contents of the manuscript, without hyperbole?

If 'No', please indicate reasons in Comments for Author box below.

Reviewer #1:

- Yes

Reviewer #2:

- No

First revision

Author response to reviewers' comments

We thank the editors and reviewers of *Biology Open* for the thoughtful and insightful comments we received and the opportunity to revise our manuscript. We have made major changes to the manuscript (as detailed below) in order to substantiate our claim that WWP1 and WBP2 interact endogenously in the human heart. For your convenience, you will find all the changes in red font or highlighted in yellow in the revised manuscript file. We believe that these changes significantly strengthen the manuscript and hope that you agree.

Reviewer 1:

Concerning the microscopy data (Fig2C): There is no quantification/analysis of the images beyond showing on the images two occurrences of signal overlap. My understanding from the text is that the immunofluorescence was performed only once. The no-primary antibody control is rather nice, although there is some staining for the WBP2 channel but I find it very difficult to reach any firm conclusion from the images presented. It would require more work to conclude that the two proteins do indeed localise to the same sub-cellular structures, yet even if they do co-localise, it does not mean that they necessarily interact in vivo. I therefore suggest to remove this statement from the text (for example, line 247: "The only interaction that was confirmed to endogenously occur in the heart was that of WWP1 and WBP2 by co-immunofluorescence."), to quantify the extent of co-localisation and to clarify whether the immunofluorescence staining was performed once or more.

>We agree that our previously presented findings were overstated. Originally, we were only able to find one unique representation of "normal human heart" as we found that most commercial sources have overlapping supply chains unless an investigator wishes to initiate a full collection protocol. During our revision period, we were able to locate and procure a limited amount of an independent human sample. While we did repeat and verify the same patterns of co-localization in both samples, we did not proceed with more rigorous co-localization analysis (using Pearson's correlation and Manders' co-occurrence) in attempt to conserve our remaining Zyagen sections for the more informative PLA analysis. As we think the data convincingly show, this assay showed signal of rare steady-state endogenous protein-protein interactions in human myocardium (Fig. 3B and 3C). We additionally bolstered this by pulling scRNA-seq data from the Heart Cell Atlas v2 which shows co-expression of WWP1 and WBP2 in ventricular myocytes as well as replicating the higher expression of WWP1 (vs. WBP2) therein (Fig. 3D). Consequently, we have left some of our original statements in the text.

The yeast two-hybrid system is used to quantify the interaction between several mutants of WWP1 and WBP2. The data appears very clear but, in my view, lacks a major control. The protein level of each of the mutant and wild-type proteins should be quantified using western-blot. If the mutants that show a reduced beta-galactosidase activity are present at a comparable level to that of the wild-type protein, then it will be possible to conclude on the strength of the interactions. This is true for the WWP1 and WBP2 mutants.

>Because such experiments do not seem to be standard for the field, we had a difficult time finding antibodies that would reliably work in our hands. In the end, we were able to detect expression of all of the mutant constructs in yeast cells. These panels have been added to Figures 4 and 5 of the revised manuscript.

Minor typo in the abstract: proteosome instead of proteasome.

>We apologize for utilizing the American spelling. We defer to the copy editor's ultimate expertise on this matter but have changed the spelling in the abstract at this time.

Reviewer 2:

The major claim that the WWP1-WBP2 interaction occurs in the human heart is not sufficiently supported by the presented data. The authors should revise these statements throughout the manuscript or provide additional experimental evidence supporting the claim.

Major Issues

1. Unsupported claim of interaction in the human heart

The manuscript repeatedly claims that the interaction between WWP1 and WBP2 occurs in the human heart. This is a major conclusion that is not supported by the current results. The only evidence presented is a widefield immunofluorescence microscopy experiment, which lacks the spatial resolution necessary to demonstrate protein-protein interactions (widefield fluorescence microscopy is ~250nm). At this stage, the data only show that WWP1 and WBP2 may be present in the same regions or compartments, not that they physically interact. Moreover, the staining patterns suggest that most of the signal is not overlapping.

To support this conclusion, the authors should modify the claim to state that the two proteins co-localize (not necessarily interact) or perform additional experiments, such as co-immunoprecipitation or proximity ligation assays, using human heart tissue to demonstrate a physical interaction.

>Thank you for your excellent suggestion. Indeed, we went ahead and did the PLA on human heart tissue with the same antibodies and were able to show PLA signal (indicating that there are areas within cardiomyocytes in which WWP1 and WBP2 are located within 40nm of one another). Please see Fig. 3B and 3C as well as the methods for further details of our analysis. Additionally, we also added scRNA-seq data from the Heart Cell Atlas v2 showing co-expression of WWP1 and WBP2 in ventricular cardiomyocytes/nuclei based on data from 25 adult donors aged 20-75 (Fig. 3D).

Specific instances where the claim should be revised include:

** Line 247: "The only interaction that was confirmed to endogenously occur in the heart was that of WWP1 and WBP2 by co-immunofluorescence."*

>With the addition of the PLA data, we have changed this sentence to read, "The only interaction that was confirmed to endogenously occur in the heart was that of WWP1 and WBP2 by PLA in a limited number of clinical samples." We feel this more accurately represents the interpretation of the data presented.

** Line 112: "Demonstrate that this interaction can occur endogenously in the heart."*

>This has been modified to state "we sought to further validate our initial findings and determine whether this interaction occurs endogenously in the heart."

Minor Issues

1. *The meaning of the phrase "Chief of the interactors" is unclear, since WBP2 ranks as the fourth hit by the number of colonies in the Y2H screen, with MTCO1 being the top interactor. The authors should explain why WBP2 was chosen for further study despite not being the top hit (e.g., biological relevance, previous literature, or domain complementarity).*

>We apologize for any confusion. The phrase was originally chosen for parsimony of word usage in the abstract. There, it has been changed to “central in the identified protein-protein interaction network” since the edge between WWP1 and WBP2 falls in the center of the STRING network. Elsewhere in the manuscript (*i.e.*, at the end of the introduction), the phrase has been changed to “Of particular interest” with further explanation.

2. *The color coding in Figure 1 uses non-standard p-value thresholds. The authors should clarify why they did not use conventional cutoffs (e.g., 0.01 or 0.001).*

>In STRING analysis, to achieve a false discovery rate of no more than 5% (*i.e.*, alpha-value < 0.05), the threshold for interaction is set to “medium confidence” of 0.400. High confidence corresponds to an FDR of 1% (*i.e.*, alpha-value <0.01). Further clarification has been provided in the text and in the legend for Fig. 1.

3. *The PY mutants should be labeled in a consistent format with the WW mutants, e.g., ΔPY1, ΔPY2-3, etc.*

>To provide more clarity, the PY mutants have now been labeled mutPY1, mutPY2, etc. in the figures. Because the D convention particularly applies to deletions whereas the PY mutants are missense variants, we felt that the “mut” prefix would be more accurate.

4. *The authors should explain why the ΔPY2-3 double mutant shows different results in the QDO medium versus the β-galactosidase assays.*

>Difficult to adequately capture succinctly is the subtle differences in the SDA growth assay on QDO medium. There are very occasional instances where the 10⁰ dilution can show some growth of the mutPY2+3 constructs in combination with WWP1 on QDO, depending on the precise amount of supplement that was added when making the media. Further, these mutants show very slow growth so if plates are left to incubate longer, sometime a little growth can be found. However, the β-galactosidase assays measure the activity of a non-essential reporter gene (*lacZ*) under the control of a different promoter. Since the expression of *lacZ* is not critical for viability, small amounts of interaction and thus reporter gene expression might be detected in this assay but not in the SDA assay where too little expression of *ADE2* or *HIS3* would be insufficient to promote growth on media lacking these supplements. Additional explanation to this effect has been provided in the text.

5. *Figure 5 could include the pairings of the individual WW mutants with the double PY mutants, particularly ΔPY2-3, which showed partial loss of interaction in the β-galactosidase assay. Since the authors already have the tools it is a simple experiment to do.*

>We agree that these experiments would have been a nice addition to the data presented in (now) Figure 6. However, in the interest of time, we prioritized the PLA data, which we believe had more of a significant value-add to our story, particularly since we did not have enough of the same lot of X-gal to do all the combinations of double mutPY constructs in combination with the single WW domain deletions and changing lots would have required re-standardization.

Second decision letter

MS ID#: bio.062347R1

MS Title: Identification and Validation of an Interaction between the E3 Ubiquitin Ligase WWP1 and the Transcriptional Co-Activator WBP2 in the Human Heart

Authors: Meaghan E. Arnold; Ymani Wright; Natalie K. Grantham; Douglas L. Pittman; Lydia E Matesic

I am happy to tell you that your manuscript has been accepted for publication in Biology Open, pending our standard publication integrity checks. It was accepted on 18th March 2026.